# Association of Anthropometrics and Body Composition with Maximal and Relative Force and Power of Kayak Stroke in Competitive Kayak Athletes

**DOI:** 10.3390/ijerph19052977

**Published:** 2022-03-03

**Authors:** Filip Kukić, Miloš Petrović, Gianpiero Greco, Stefania Cataldi, Francesco Fischetti

**Affiliations:** 1Police Sports Education Center, Abu Dhabi Police, Abu Dhabi 253, United Arab Emirates; filip.kukic@gmail.com; 2Faculty of Sport and Physical Education, University of Belgrade, 11030 Belgrade, Serbia; mikac1989@yahoo.com; 3Department of Basic Medical Sciences, Neuroscience and Sense Organs, University of Study of Bari, 70124 Bari, Italy; gianpierogreco.phd@yahoo.com (G.G.); francesco.fischetti@uniba.it (F.F.)

**Keywords:** sprint kayak, canoe sport, strength, performance modeling, skeletal muscle mass

## Abstract

This study determined to what degree anthropometrics and body composition are associated with force and power outputs of a single-stroke kayak test (SSKT). Body height (BH), sitting height, biacromial distance, arm span, stroke length, body mass index (BMI), percent of skeletal muscle mass (PSMM), skeletal muscle mass index (SMMI), percent body fat (PBF) and maximal and relative force (SSKT_Fmax_ and SSKT_Frel_) and power (SSKT_Pmax_ and SSKT_Prel_) of the SSKT were assessed in 21 male kayak competitors, including sprint specialists and long-distance specialists. Correlation analysis established the association (*p* < 0.05) between SSKT_Fmax_ and BM (r = 0.511), BMI (r = 0.495) and SMMI (r = 0.530). A significant correlation (*p* < 0.05) also occurred between SSKT_Pmax_ and BMI (r = 0.471) and SMMI (r = 0.489). Regression analysis determined a significant association of the anthropometrics–body composition model of kayakers with SSKT_Fmax_ (R^2^ = 0.790), SSKTF_Rel_ (R^2^ = 0.748), SSKT_Pmax_ (R^2^ = 0.676) and SSKT_Prel_ (R^2^ = 0.625). A longer and wider upper body supported by higher amounts of skeletal muscle mass per square of body size provides higher force outcomes in a complex single-handed SSKT, while the PSMM provides higher outcomes in SSKT_Pmax_.

## 1. Introduction

Sprint kayak is an Olympic sport where a defined sequence of movements repeats over the time needed for the race to be finished. Kayak athletes compete in a seated position with their legs extended anteriorly using a double-blade paddle to propel the water [1,2]. Therefore, they apply propulsion forces on each side of the boat [3]. In order to transfer the forces of hip and upper body rotation, the muscles that pull and abduct the lower hand and the muscles that fix and push the upper hand must produce forces that overcome the drag forces. Moreover, these muscles do not just passively transfer the forces of the rotation but rather actively engage by pulling (lower arm) and pushing (upper arm) the paddle [3,4]. Indeed, the muscular strength and endurance of the legs are of great importance in supporting and generating the upper body rotation. However, the strength and power training of the upper body is typically prioritized over the strength and power training of the legs because of its more dominant involvement in stroke performance. Therefore, kayak athletes typically possess very strong upper body and hand muscles such as the pectoralis, latissimus, deltoideus, biceps and triceps muscles, supported by sufficiently strong legs.

Paddling efficacy could be defined as the time taken to complete the race, whereby to maximize the average velocity of the kayak, the athlete needs to generate high average power during each stroke and large average forces on the paddle blade during the propulsion phase [2,5]. Thus, identifying the attributes of kayak athletes such as body composition, muscular force, strength and power is of high importance [6]. In general, muscular force is the only force directly controlled by the central nervous system [7], which makes skeletal muscle mass an important component of kayak athletes’ body composition. On the other hand, to move the boat forward, the athletes must generate enough propulsive force to overcome the hydrodynamic and aerodynamic drag forces placed on the boat and athlete [2,5,8]. Given that the kayak athlete is fixed in the kayak via the seat and foot bar, increased body fatness may increase the drag force and reduce the paddling efficacy [2,4].

Considering this, the body composition of kayak athletes may be of importance to consider for long-, medium- and short-term training planning [6]. Indeed, anthropometric dimensions such as body height (BH), seated height and arm span also affect the physical performance of athletes [9,10,11,12]. However, the difference between body composition and anthropometric dimensions is that anthropometrics is mostly genetically predetermined, while body composition is the reflection of adaptation to training and nutrition. Considering this, body composition could be part of the screening process, while anthropometric measures are often used in the selection process and talent acquisition [10,12,13].

Several studies investigated the morphological profiles of flat-water kayak athletes [10,13,14,15]. However, only few studies investigated the association of kayak athletes’ morphology with kayak stroke performance [15,16]. This is partially due to difficulty in assessing isolated stroke performance. For this reason, Petrović et al. [17,18] recently developed a kayak stroke test sensitive enough to detect differences between competitors in different disciplines. This test is conducted on a custom-made railing system, which, after a kayaker performs the kayak stroke, rolls forward until the rolling stops. This system provides the rolling velocity and distance, which mimics the on-water movement of the boat–athlete system. Having a sensitive tool for the assessment of muscle characteristics involved in kayak strokes allows the investigation of the association of morphological characteristics with single-hand kayak stroke performance. Thus, the aim of this study was to investigate the association of anthropometric and body composition characteristics with the force and power outputs produced during a single kayak stroke. In addition, this study determined the prediction value of anthropometric and body composition measures in the evaluation of single-stroke performance. We hypothesized that the anthropometrics and body composition would be significantly associated with the force and power outputs of a single kayak stroke, and that the anthropometrics and body composition would be significant predictors of force and power outputs of kayak strokes.

## 2. Materials and Methods

### 2.1. Experimental Approach to the Problem

A cross-sectional study was carried out to investigate the association of body composition with the force and power outputs of a single-stroke kayak test (SKST). Participants came to the laboratory on two occasions separated by 48–96 h.

On the first occasion, participants conducted the assessment of anthropometrics and body composition. The participants were asked not to eat or drink anything for a minimum of 3 h before the test because of the body composition test. However, they were provided with drinking water and energy bars at the test site in case they needed to replenish after the body composition assessment.

On the second occasion, participants performed the SSKT. Before commencing the SSKT, participants completed the warm-up consisting of five 5 min sessions of pedaling on a kayak ergometer at a self-regulated low to moderate intensity (i.e., 50–80% of their ergometer race pace), which was followed by 3–5 maximal accelerations of 5–10 s and 5 min of dynamic stretching of the muscles involved in the tests. Testing sessions were held during the afternoon from 13:00 to 17:00, and participants were instructed not to perform any strenuous exercise the day before testing.

### 2.2. Participants

The sample consisted of 21 male kayakers: 7 sprint specialists and 14 long-distance specialists (age = 21.4 ± 3.14 years [17,18,19,20,21,22,23,24,25,26,27,28,29]; body height = 180.78 ± 6.26 cm; and body mass [BM] = 80.08 ± 7.11 kg). All participants were active competitors at either national (*n* = 12) or international level (*n* = 9, e.g., Serbian national team), with a minimum competing history of 5 years and a training frequency of a minimum of four times per week (4–12). Two participants were 2018 World Champions and 2016 Olympic vice-champions, and two were medalists at the European and World Championships. None of them had recent injuries or musculoskeletal pain that could negatively affect the results of the study. Participants and coaches were thoroughly briefed about the tests and informed about the aim and purpose of the study. All participants signed written informed consent. One participant was 17 years old, so their parent or legal guardian gave written consent. The study was approved by the ethical board of the Faculty of Sport and Physical Education, University of Belgrade (02-282/12-1, 20 February 2018), and was carried out in accordance with the conditions of the Declaration of Helsinki.

### 2.3. Anthropometrics and Body Composition

Anthropometric measurements were assessed using an anthropometer (SECA-792, Hamburg, Germany) following procedures used in previous studies on kayak competitors [19,20]. When sitting height was measured, the participant was seated on a measurement box with their back and buttocks touching the backboard of the stadiometer, knees directed straight ahead and arms and hands resting at their side. For both standing and sitting height measurements, the head was in the horizontal plane. Arm span was measured from the tip of the middle finger on one hand to the tip of the middle finger on the other hand with the individual standing against the wall with both arms abducted to 90°, the elbows and wrists extended and the palms facing directly forward. The between-acromial distance was measured by a sliding caliper (Vogel, Kevelaer, Germany) while the participant was standing upright with their arms hanging loosely at their sides [21]. All anthropometric characteristics were measured three times with 1 mm reading accuracy, and the average of three measurements was used for the analysis. To ensure the lowest technical error of the measurement, a highly skilled medical professional (i.e., professor of human anatomy from the Department of Sports Medicine) performed the anthropometric measurements.

Body composition measurement procedures were conducted using an 8-channel bioelectric impedance device (InBody 720: Biospace Co., Ltd., Seoul, Korea), which was shown to be very reliable with an ICC = 0.97 [22]. The assessment was conducted as previously reported in detail [23,24,25]. In short, participants were measured in underwear, barefooted, and had all metal, plastic and magnetic accessories removed. They stood on the device and on the metal spots designated for their feet and held the handles for the designated spots. The outcome measures from this device that were relevant to this study were body mass (BM), body fat mass and skeletal muscle mass, which were later used to calculate the body composition index that reflects the relative quantity of a certain body tissue or the quantity of the tissue relative to body size [25,26]. Tissue quantity was examined for body fatness by percent body fat (PBF) and body muscularity by the percent of skeletal muscle mass (PSMM). The quality of muscular tissue was examined using a skeletal muscle mass index (SMMI).

### 2.4. Single-Stroke Kayak Force and Power

The SSKT was performed on a customized railing system recently described in detail [17]. In short, two rigid cables attached the roller device to the custom-made lat machine, depending on the magnitude of the load. A short cable (total length 390 cm and pull-out length 155 cm) was used for the heavy loads, transmitting the real force of fourteen pieces of 5 kg weight discs (>10 kg). A longer cable (total length 1210 cm and pull-out length 875 cm) was used for light loads, transmitting six times lower force (≤10 kg). The railing system was 20 cm wide and 11 m long, while the dimensions of the roller device were 120 × 25 × 25 cm, and it weighed 12 kg. Participants sat in a kayak seat connected to the roller device in the typical position for kayak athletes and held the simulation of the kayak paddle that was connected to a stable non-movable bar that was part of the railing system (Appendix A). As participants performed the stroke, the roller device slid along the railing system. Therefore, the muscle forces had to overcome the weight of the participant–roller system and friction force in order to move the system forward. A right stroke was performed in this study, where the right arm was pulling, and the left arm was pushing. Participants were instructed to perform the stroke as forceful and strong as possible, like they would do in a kayak. The railing system was 11 m long, thus allowing the participants to perform the stroke without the fear of how they will stop. The signal to start was “Ready, Go!”. None of the participants reported discomfort in the starting position. They were allowed two to three trials to familiarize themselves with the device and the test and three to four additional trials to test their 1RM. Given that participants were professional competitors, with extensive training history, and that the SSKT mimicked on-water and off-water paddling, participants did not need a session dedicated to familiarization.

The length of the stroke was defined as the distance between the starting position (e.g., maximal forward reach of the pulling hand—the end of the paddle simulation aligned with the feet) and the end position (e.g., the end of the paddle simulation aligned with the trochanter major). The 1RM was the maximum weight a subject could overcome from the starting to the end position without deviating from the proper kayak stroke technique. Average values of force and velocity were collected with a linear position transducer (LPT) (Real Power Pro Globus, Codogne, Italy), which was previously validated [27]. The LPT was attached perpendicularly to the first weight plate of the lat machine and sampled the displacement–time data at a frequency of 1000 Hz. The velocity value recorded by the LPT was multiplied by a coefficient of transmission of 5.935 when the long cable was used. Furthermore, the coefficient of friction force (μ = 0.086) was measured before testing, and it was considered to correct force computations.

#### Force and Power Variables

Four variables were used from the SSKT: the maximal F and P (SSKT_Fmax_ and SSKT_Pmax_) and the relative F and P (SSKT_Frel_ and SSKT_Prel_) of the SSKT. SSKT_Fmax_ is the maximal force produced during the stroke. The maximal power of SSKT_Pmax_ was modeled from the intercept P_max_ = *F*_0_*∙V*_0_*/*4, where *V*_0_ was calculated from the velocity intercept [*V_0_ = F*_0_*/a*], while *F*_0_ represents the force intercept, and *a* is the slope of the F–V relationship (F–V slope) [7,17]. F_max_ and Pmax were relativized to body mass powered to 2/3: [Frel = Fmax * (body mass^2/3^)] [28,29].

### 2.5. Statistical Analyses 

The descriptive statistics were calculated for the mean, standard deviation (Std. Dev.), minimum (Min.) and maximum (Max.). The Shapiro–Wilk test was used to identify the normality of the data distribution, and it revealed that data in all variables were normally distributed. Pearson’s correlation coefficient was used for the variables that had a normal distribution. A backward linear regression was used to determine the best anthropometrics and body composition predictors of SSKT_Fmax_, SSKT_Frel_, SSKT_Pmax_ and SSKT_Prel_, with the lowest standard error of the estimate. The significance level was set to *p* < 0.05, and type I error was controlled by adding 95% confidence intervals. The magnitude of correlations was defined as: small = 0.20–0.49, medium = 0.50–0.79 and large ≥ 0.80, and the magnitude of regression R^2^ was defined as: small = 0.04–0.24, medium = 0.25–0.63 and large > 0.64 [30].

## 3. Results

The descriptive statistics for the mean, standard deviation, minimum and maximum are shown in Table 1. The minimum and maximum values of PBF and PSMM indicate that the sample was homogenous by relative tissue volumes as all participants were highly muscular with low levels of body fat.

Considering the anthropometric characteristics, only BM correlated with SSKT_Fmax_ (Table 2), while BMI and SMMI correlated significantly with SSKTFmax and SSKT_Pmax_. However, PBF and PSMM did not correlate with any characteristic of the SSKT. In addition, the correlation showed large coefficients of association of SSKT_Pmax_ with SSKT_Fmax_ (r = 0.837, *p* < 0.001) and SSKT_Frel_ (r = 0.822, *p* < 0.001), and of SSKT_Prel_ with SSKTFmax (r = 0.744, *p* < 0.001) and SSKTFrel (r = 0.838, *p* < 0.001).

Linear regression analysis determined significant prediction models (Figure 1) for SSKT_Fmax_ (ajd.R^2^ = 0.649, Std. Error of the Est. = 56.56, F = 5.631, *p* = 0.004), SSKT_Frel_ (ajd.R^2^ = 0.581, Std. Error of the Est. = 2.97, F = 4.460, *p* = 0.010), SSKT_Pmax_ (ajd.R^2^ = 0.502, Std. Error of the Est. = 41.16, F = 3.874, *p* = 0.017) and SSKT_Prel_ (ajd.R^2^ = 0.424, Std. Error of the Est. = 2.294, F = 3.100, *p* = 0.037). The magnitude of association was large in all four prediction models. 

The determination of coefficients and their 95% confidence intervals (lower bound–upper bound) for each prediction model is shown in Table 3. Considering individual indicators, BH, SH, BAD and SL were the most significant anthropometric measures, while SMMI and PBF were the most significant body composition measures, in predicting SSKT_Fmax_ and SSKT_Frel_. BH had a negative effect, while SH had a positive effect, on force outputs of kayak strokes. Considering power outputs, BH was the only individually significant predictor of SSKT_Pmax_ and SSKT_Prel_, while PSMM and PBF were significantly associated with only SSKT_Prel_. Considering the 95% confidence intervals, SMMI and BMI varied the most in relation to force outputs, while PSMM and PBF varied the most in relation to power outputs (Figure 2). Variations in anthropometric characteristics were larger in association with power than they were in association with force outputs.

## 4. Discussion

This study investigated the association of anthropometric and body composition characteristics with force and power outputs of an SSKT and determined the prediction value of anthropometric and body composition indicators. The main findings of this research revealed that only SMMI and BMI correlated with SSKT_Fmax_ and SSKT_Pmax_, while there was no other significant association of anthropometrics and body composition measures with SSKT outputs. However, SSKT_Fmax_ and SSKT_Frel_ strongly correlated with power outputs of the SSKT. Therefore, the higher the muscle quality, the higher the force and power of outputs of muscle contraction. The regression analysis determined significant prediction models for all measured SSKT outcomes, whereby BH, SH, BAD, SMMI and PBF were significant predictors in the SSKT_Fmax_ and SSKT_Frel_ prediction models; BH, BAD, PSMM and PBF were significant predictors in the SSKT_Pmax_ prediction model, while BH was the only significant predictor in the SSKT_Prel_ prediction model. The negative prediction by BH and the positive prediction by SH, BAD, SMMI and PSMM suggest that the upper body of kayakers tends to be longer, with wider shoulders and more muscular strength for better force production and power outcomes. This results in the upper body strength needed to move the athlete–kayak system through the water.

When investigating the influence of upper body strength on sprint kayak performance in elite kayak athletes, McKean and Burkett [31] found that significant improvements over a two-year period in 1RM pull-up (29.9%, *p* = 0.002) and 1RM bench press (38.1%, *p* = 0.006) were followed by improvements in race time at 200 m (−3.1%), 500 m (−2.5%) and 1000 m (−3.1%). Moreover, the authors reported a significant moderate to large correlation between 1RMs of these two exercises and race time, whereby the strength of the relationship increased as the distance decreased. This may be related to the larger stroke force and power produced by short-distance competitors compared to long-distance competitors, as shown in the study by Petrovic et al. [17]. Skeletal muscle is a highly malleable tissue capable of adapting in response to exercise, which may vary between endurance and resistance exercises, and the volume, intensity and frequency of exercise [32]. Therefore, training-induced adaptations in skeletal muscles are different between athletes from different sports or different specializations within their sport [26,33,34,35,36]. Methentis et al. [33] conducted a fiber-level analysis of the difference in muscle morphology of differently trained individuals and whether muscle morphology is associated with the rate of force development. The authors found that power-trained athletes had a higher percentage of type II muscle fibers within the cross-sectional area of the vastus lateralis than the endurance-trained athletes. Moreover, they reported a very large positive correlation between the percent of type II muscle fibers and the rate of force development, and a large negative correlation between the rate of force development and type I muscle fibers. Considering that the sample for this study included power-related (200 m) and endurance-related (500 m and 1000 m) kayak competitors, it could be argued that the power-related competitors possess a muscle morphology that is better suited for force and power development. Additionally, of note is that the sample from the current study included medalists from the World Championships in both the 200 m and 1000 m disciplines; thus, highly specific adaptations to training are certain, additionally reinforcing the aforementioned notion. In that regard, given that SMMI and PSMM are significant predictors of force and power outputs of the SSKT, it seems that discipline-specific SMMI and PSMM models of elite kayak athletes could be defined in the future.

This was further reflected in the regression analysis, whereby kayakers who were more forceful and powerful in the SSKT tended to have more muscular bodies. This could be expected based on previous studies that defined the effect of body size on muscular power and force [24,25]. However, it provides evidence that the adaptation of body composition and selection of competitors by anthropometrics transfer into SSKT performance. This could be observed when analyzing the 95% confidence intervals for individual measures of anthropometrics and body composition. The intervals were larger in body composition, providing a larger margin for training effects, while intervals in anthropometrics were smaller, suggesting stronger effects of selection and talent identification processes. Therefore, body composition measures such as SMMI, PSMM and PBF are among the main factors for planning and programming specific and non-specific training for the development of mechanical characteristics of kayak strokes. This could also be a partial explanation for why PBF entered the regression model because international-level competitors were within the last few weeks of the pre-competition period and, accordingly, were expected to be leaner (e.g., have lower PBF) than those who were national-level competitors. In addition, during the selection process for the national team, in the pre-competition period or early competition period, other non-national team competitors could replace competitors from the national team in case they perform better. In that regard, screening competitors’ PBF could provide early signs to re-evaluate training plans and programs, nutrition and health.

### Limitations

This study is not without limitations. The sample of international-level competitors of each specialty as well as the sample of non-international-level competitors of each specialty could be larger in order to develop a stronger model of evaluation of mechanical outputs of single-stroke kayak performance via anthropometrics and body composition. Moreover, the sample did not include different age categories which would allow defining models that competitors should reach at each age in order to be able to produce force and power in kayak strokes that are needed for the international level of competition. The sample did not include females, and the obtained results apply only to male kayak competitors. For higher validity, the force and power of a single kayak stroke should be obtained in the water.

## 5. Conclusions

This study determined the degree to which anthropometrics and body composition correspond to mechanical characteristics of kayak strokes. The results suggest that competitive kayakers tend to have longer, wider and more muscular upper bodies for better force and power outputs of their kayak stroke. This information indicates that the adaptation to training could be followed by assessment of body composition characteristics such as SMMI, PSMM and PBF, while anthropometrics may be of importance in selection and talent identification processes. PSMM and SMMI are indicators of acute and cumulative training effects on the muscularity of kayakers. In contrast, PBF could indicate acute changes (e.g., increase in body fatness) that suggest possible issues in the training plan or the competitor’s behavior, nutrition or health. This study provides a practical approach for better precision in modeling kayakers’ bodily characteristics for better kayak stroke mechanical outputs. For sports that are hard to be evaluated in real conditions such as sprint kayak, this information could be very useful for planning and monitoring off-water and on-water training processes. It is easily feasible and provides a useful cross-sectional insight into the athletes’ preparedness. If utilized regularly during the season, it would establish clear trends that could be used for permanent follow-up of individual athletes in relation to their specialty. In addition, the model could be developed for different age categories, which would allow a more precise utilization of training technology.

## Figures and Tables

**Figure 1 ijerph-19-02977-f001:**
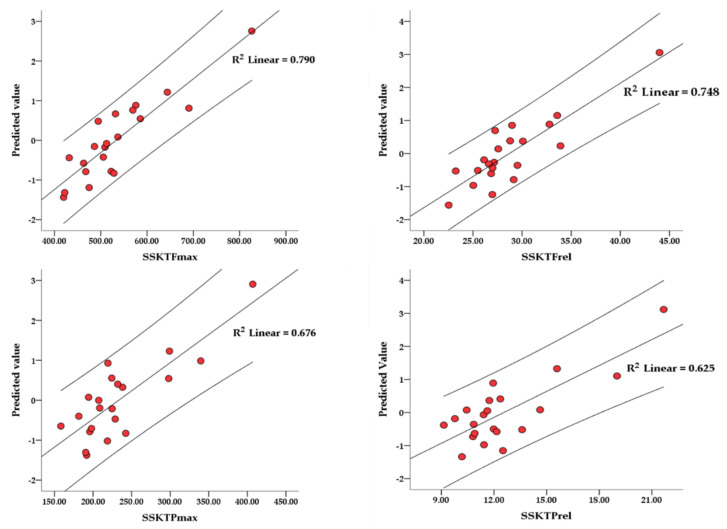
The regression coefficient with the lowest standard error of the estimate for SSKT_Fmax_, SSKT_Frel_, SSKT_Pmax_ and SSKT_Prel_. The prediction value was calculated via coefficients obtained from the regression analysis.

**Figure 2 ijerph-19-02977-f002:**
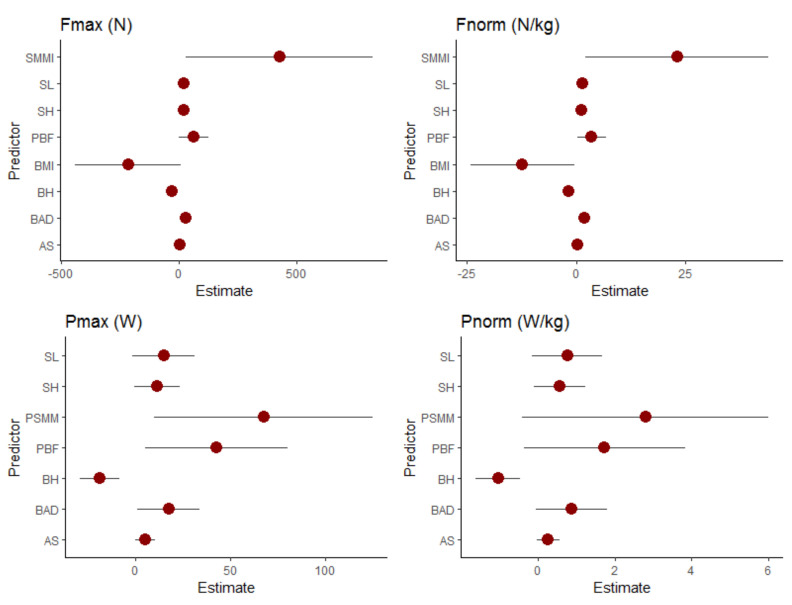
Prediction coefficients. Note: BH—body height; SH—sitting height; BAD—biacromial distance; AS—arm span; SL—stroke length; BMI—body mass index; PBF—percent body fat; SMMI—skeletal muscle mass index; PSMM—percent of skeletal muscle mass.

**Table 1 ijerph-19-02977-t001:** Descriptive statistics.

Variables	Mean	Std. Dev.	Min.	Max.	SWT
Age (years)	21.67	3.14	17.00	29.00	0.108
BH (cm)	180.78	6.26	167.80	190.60	0.688
BM (kg)	80.08	7.11	64.50	95.40	0.895
Sitting height (cm)	95.88	3.20	87.40	101.50	0.502
Between-acromial distance (cm)	43.1381	1.63	40.70	47.00	0.600
Arm span (cm)	188.4286	7.85	176.00	199.50	0.074
Stroke length (cm)	97.4286	2.86	92.00	102.00	0.854
Body mass index (kg/m²)	24.49	1.62	22.83	28.13	0.066
Percent body fat (%)	9.56	3.44	3.05	15.39	0.779
Percent of skeletal muscle mass (%)	51.99	2.24	48.15	55.89	0.723
Skeletal muscle mass index (kg/m²)	12.73	0.98	11.00	14.50	0.807
Single-stroke kayak Test_Fmax_ (N)	533.27	95.52	419.84	826.34	0.233
Single-stroke kayak Test_Frel_ (N/kg^2/3^)	28.70	4.59	22.53	44.00	0.143
Single-stroke kayak Test_Pmax_ (W)	233.26	58.30	158.35	406.98	0.114
Single-stroke kayak Test_Prel_ (W/kg^2/3^)	12.56	3.02	9.14	21.67	0.081

Note: SWT, Shapiro–Wilk test.

**Table 2 ijerph-19-02977-t002:** Correlation coefficients for association of anthropometrics and body composition with kayak stroke characteristics.

	BH	BM	SH	BAD	AS	SL	BMI	PBF	PSMM	SMMI
SSKT_Fmax_	0.197	0.511 *	0.430	0.231	0.263	0.322	0.495 *	−0.090	0.189	0.530 *
SSKT_Frel_	−0.130	0.069	0.132	0.107	0.008	0.066	0.241	−0.029	0.091	0.253
SSKT_Pmax_	−0.057	0.286	0.183	0.256	0.077	0.066	0.471 *	−0.084	0.167	0.489 *
SSKT_Prel_	−0.227	0.051	0.014	0.190	−0.057	−0.075	0.332	−0.052	0.113	0.337

Note: * Correlation is significant at the 0.05 level. BH—body height; BM—body mass; SH—sitting height; BAD—biacromial distance; AS—arm span; SL—stroke length; BMI—body mass index; PBF—percent body fat; PSMM—percent of skeletal muscle mass; SMMI—skeletal muscle mass index.

**Table 3 ijerph-19-02977-t003:** The determination of coefficients that entered the prediction models.

Model		Unstandardized Coefficients		Standardized Coefficients	t	Sig.	95.0% Confidence Interval for B	
		B	Std. Error	Beta			Lower Bound	Upper Bound
SSKT_Fmax_ (N)	(Constant)	−1958.73	589.69		−3.32	0.006	−3243.55	−673.92
	BH	−29.55	7.25	−1.94	−4.08	0.002	−45.34	−13.76
	SH	20.87	7.55	0.70	2.77	0.017	4.43	37.31
	BAD	32.16	11.16	0.55	2.88	0.014	7.84	56.48
	AS	6.65	3.46	0.55	1.92	0.079	−0.89	14.20
	SL	24.29	10.18	0.73	2.39	0.034	2.11	46.48
	BMI	−213.83	102.66	−3.63	−2.08	0.059	−437.52	9.85
	PBF	64.71	28.96	2.33	2.23	0.045	1.60	127.81
	SMMI	427.56	182.36	4.39	2.34	0.037	30.24	824.88
SSKT_Frel_ (N/kg^2/3^)	(Constant)	−49.37	31.02		−1.59	0.137	−116.95	18.21
	BH	−1.80	0.38	−2.46	−4.72	0.000	−2.63	−0.97
	SH	1.12	0.40	0.78	2.81	0.016	0.25	1.98
	BAD	1.77	0.59	0.63	3.02	0.011	0.49	3.05
	AS	0.36	0.18	0.62	2.00	0.069	−0.03	0.76
	SL	1.29	0.54	0.80	2.41	0.033	0.13	2.46
	BMI	−12.29	5.40	−4.34	−2.28	0.042	−24.06	−0.53
	PBF	3.51	1.52	2.63	2.30	0.040	0.19	6.82
	SMMI	23.02	9.59	4.91	2.40	0.034	2.12	43.92
SSKT_Pmax_ (W)	(Constant)	−4536.72	1557.72		−2.91	0.012	−7901.97	−1171.47
	BH	−18.84	4.81	−2.02	−3.91	0.002	−29.24	−8.44
	SH	11.31	5.56	0.62	2.03	0.063	−0.70	23.33
	BAD	17.52	7.60	0.49	2.30	0.038	1.09	33.95
	AS	5.16	2.47	0.69	2.09	0.057	−0.18	10.49
	SL	14.81	7.57	0.73	1.96	0.072	−1.55	31.17
	PBF	42.62	17.47	2.52	2.44	0.030	4.86	80.37
	PSMM	67.57	26.67	2.59	2.53	0.025	9.96	125.18
SSKT_Prel_ (W/kg^2/3^)	(Constant)	−176.51	86.84		−2.03	0.063	−364.13	11.10
	BH	−1.04	0.27	−2.15	−3.87	0.002	−1.62	−0.46
	SH	0.56	0.31	0.59	1.80	0.094	−0.11	1.23
	BAD	0.87	0.42	0.47	2.06	0.060	−0.04	1.79
	AS	0.27	0.14	0.69	1.93	0.076	−0.03	0.56
	SL	0.76	0.42	0.72	1.79	0.096	−0.15	1.67
	PBF	1.73	0.97	1.97	1.78	0.099	−0.37	3.84
	PSMM	2.79	1.49	2.07	1.88	0.083	−0.42	6.01

BH—body height; SH—sitting height; BAD—biacromial distance; AS—arm span; SL—stroke length; BMI—body mass index; PBF—percent body fat; SMMI—skeletal muscle mass index; PSMM—percent of skeletal muscle mass; SSKT—single-stroke kayak test.

## Data Availability

The data presented in this study are available on request from the first author. The data are not publicly available due to privacy.

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
