# Peer review of "Association of Anthropometrics and Body Composition with Maximal and Relative Force and Power of Kayak Stroke in Competitive Kayak Athletes"

_ijerph, 2022, doi:10.3390/ijerph19052977_

Round 1

Reviewer 1 Report

This study suggested interesting results and experimental suggestions.

Also, this paper is well written with logical flow.

However, I found some minor limitations and suggest revision related these issues.

This paper deserves to be published after minor revisions.

30-39 Need a references to support your statement.

48-50 You need references to support your statement or reposition the phrase as a hypothesis.

73-85 I do not clearly see the procedures timeline clear. A figure with the time-line may help on that.

102 The brand and city of the material used are not mentioned.

102-112 The methodology used is not referenced.

136 Figure 1 is cited implying that it explains the performance of the test, but this figure corresponds to the results.

179 BPRFmax is referenced in Table 1, but I cannot locate it in that Table.

187-188 Table 2 Values of p<0.01 that are referenced in the Note are not reflected.

Reviewer 2 Report

Introduction

Line 32-38: The rational about the importance of the upper-body must be expanded. The way it is it seems that lower-body, despite non producing direct propulsive force, it´s not important for the overall force generating.

Line 39: reference needed.

Line 64: in what this consists of?

Line 70: please add a hypothesis

Methods

Line 74: This SKST test must be written first (abstract doesn’t count)

Line 79: what is low to moderate? Specify what % of what.

Lines 102 – 112: please add references whenever suitable. How distances were measured? Manually or by photogrammetry? Please explain. If manually, ICC must be included.

Did the authors consider to collect flexibility variables?

Lines 126-144: were the participants’ familiarized with this protocol?

Line 164: with a sample under 30 participants, the standard test to check normality is the Shapiro-Wilk.

Line167-168: if you used parametric and non-parametric correlations, for the regression did you plot all variables (i.e., those which were normally and non-normally distributed) together?

Results

Table 1  - I think you should test normality with Shapiro-Wilk and see if the variables are normally distributed. Moreover, I think you should re-write the stats section and mention that all variables were normally distributed (if Shapiro-Wilk test reveals to be equal to Kolmogorov-Smirnov). Only age wasn’t normally distributed. And you didn´t even use it for the correlation or regression. The way it is, it can be misleading. You don’t even include in Table 1 the BPRFmax, which you claim to be non-normally distributed, and I don’t know what this variable refers to!

I thought that SSKT was the dependent variable Only one). But by reading the results, there are four dependent variables that were driven from this test (Fmax, Frel, Pmax, and Prel). This should be clearly explained in the methods section.

Figure 1 is the linear regression between the SSKT variables and what exactly? By the stats sections, I thought authors were only computing data shown in table 3. Also, what indicates Figure 2? Please clearly explain this in the stats section.

Table 3 – please include only (for each dependent variable) the significant predictors.

In the stats and results sections I see nothing about a comparison between sprint and long-distance specialists. By reading the abstract and sample in the methods section, I thought that descriptive stats would be presented at least by distance specialty (i.e., sprint vs long-distance). Conversely, the authors plotted all athletes together and analyzed the correlations between anthropometrics and muscle strength parameters with the SSKT. I think this data should be presented separately: descriptive of sprinters, descriptive of long-distance. Correlations of sprinter variables, correlations of long-distance variables. Regression models for sprinters and regression models for long-distance. Comparison between specialists can also be performed.

The only way to not show data like this, is if authors previously run a t-test and acknowledge that there are non-significant differences in anthropometrics, muscle strength and SSKT between sprinters and long-distance specialists.

Round 2

Reviewer 2 Report

The authors improved the manuscript. KST should be changed in table 1 to Shapiro-Wilk test. Other typos must be edited if necessary.

Author Response

Dear Reviewer,

We thank you for your time in making suggestions for improving our manuscript.

Below is our response to your suggestion.

Reviewer: The authors improved the manuscript. KST should be changed in table 1 to Shapiro-Wilk test. Other typos must be edited if necessary.

Reply: Done. Thank you so much!